# Image to Sphere: Learning Equivariant Features for Efficient Pose Prediction

**David M. Klee, Ondrej Biza, Robert Platt, Robin Walters**
Northeastern University
{klee.d, biza.o, r.platt, r.walters}@northeastern.edu

## Abstract

Predicting the pose of objects from a single image is an important but difficult computer vision problem. Methods that predict a single point estimate do not predict the pose of objects with symmetries well and cannot represent uncertainty. Alternatively, some works predict a distribution over orientations in $SO(3)$. However, training such models can be computation- and sample-inefficient. Instead, we propose a novel mapping of features from the image domain to the 3D rotation manifold. Our method then leverages $SO(3)$ equivariant layers, which are more sample efficient, and outputs a distribution over rotations that can be sampled at arbitrary resolution. We demonstrate the effectiveness of our method at object orientation prediction, and achieve state-of-the-art performance on the popular PASCAL3D+ dataset. Moreover, we show that our method can model complex object symmetries, without any modifications to the parameters or loss function. Code is available at https://dmklee.github.io/image2sphere.

## 1 Introduction

Determining the pose of an object from an image is a challenging problem with important applications in artificial reality, robotics, and autonomous vehicles. Traditionally, pose estimation has been approached as a point regression problem, minimizing the error to a single ground truth 3D rotation. In this way, object symmetries are manually disambiguated using domain knowledge (Xiang et al., 2018) and uncertainty is not accounted for. This approach to pose estimation cannot scale to the open-world setting where we wish to reason about uncertainty from sensor noise or occlusions and model novel objects with unknown symmetries.

Recent work has instead attempted to learn a distribution over poses. Single rotation labels can be modeled as random samples from the distribution over object symmetries, which removes the need for injecting domain knowledge. For instance, a table with front-back symmetry presents a challenge for single pose regression methods, but can be effectively modeled with a bimodal distribution. The drawback to learning distributions over the large space of 3D rotations is that it requires lots of data, especially when modeling hundreds of instances across multiple object categories.

This poor data efficiency can be improved by constraining the weights to encode symmetries present in the problem (Cohen & Welling, 2016). The pose prediction problem exhibits 3D rotational symmetry, e.g. the $SO(3)$ abstract group. That is, if we change the canonical reference frame of an object, the predictions of our model should transform correspondingly. For certain input modalities, such as point clouds or 3D camera images, the symmetry group acts directly on the input data via 3D rotation matrices. Thus, many networks exploit the symmetry with end-to-end $SO(3)$ equivariance to achieve sample efficient pose estimation. However, achieving 3D rotation equivariance in a network trained on 2D images is less explored.

Thus, we present Image2Sphere, I2S, a novel method that learns $SO(3)$-equivariant features to represent distributions over 3D rotations. Features extracted by a convolutional network are projected from image space onto the half 2-sphere. Then, spherical convolution is performed on the features with a learned filter over the entire 2-sphere, resulting in a signal that is equivariant to 3D rotations. A final $SO(3)$ group convolution operation produces a probability distribution over $SO(3)$ parameterized in the Fourier domain. Our method can be trained to accurately predict object orientation and correctly express ambiguous orientations for objects with symmetries not specified at training time.

I2S achieves state-of-the-art performance on the PASCAL3D+ pose estimation dataset, and outperforms all baselines on the ModelNet10-SO(3) dataset. We demonstrate that our proposed architecture for learning SO(3) equivariant features from images empirically outperforms a variety of sensible, alternative approaches. In addition, we use the diagnostic SYMSOL datasets to show that our approach is more expressive than methods using parametric families of multi-modal distributions at representing complex object symmetries.

**Contributions:**

- We propose a novel hybrid architecture that uses non-equivariant layers to learn SO(3)-equivariant features which are further processed by equivariant layers.
- Our method uses the Fourier basis of SO(3) to more efficiently represent detailed distributions over pose than other methods.
- We empirically demonstrate our method is able to describe ambiguities in pose due to partial observability or object symmmetry unlike point estimate methods.
- I2S achieves SOTA performance on PASCAL3D+, a challenging pose estimation benchmark using real-world images.

## 2    RELATED WORK

**6D Pose Estimation**    Predicting the 6D pose (e.g. 3D position and orientation) of objects in an image has important applications in fields like robotics (Tremblay et al., 2018), autonomous vehicles (Geiger et al., 2012), and microscopy (Levy et al., 2022). Most popular methods use deep convolutional networks, which are robust to occlusions and can handle multi-object scenes with a segmentation module (He et al., 2017). Convolutional networks have been trained using a variety of output formulations. Xiang et al. (2018) regresses the 3D bounding box of the object in pixel space, while He et al. (2020) predicts 3D keypoints on the object with which the pose can be extracted. Another line of work (Wang et al., 2019; Li et al., 2019; Zakharov et al., 2019) outputs a dense representation of the object's coordinate space. Most of these methods are benchmarked on datasets with limited number of object instances (Hinterstoisser et al., 2011; Xiang et al., 2018). In contrast, our method is evaluated on datasets that have hundreds of object instances or novel instances in the test set. Moreover, our method makes minimal assumptions about the labels, requiring only a 3D rotation matrix per image regardless of underlying object symmetry.

**Rotation Equivariance**    Symmetries present in data can be preserved using equivariant neural networks to improve performance and sample efficiency. For the symmetry group of 3D rotations, SO(3), a number of equivariant models have been proposed. Chen et al. (2021) and Fuchs et al. (2020) introduce networks to process point cloud data with equivariance to the discrete icosahedral group and continuous SO(3) group, respectively. Esteves et al. (2019b) combines images from structured viewpoints and then performs discrete group convolution to classify shapes. Cohen et al. (2018a) introduces spherical convolution to process signals that live on the sphere, such as images from 3D cameras. However, these methods are restricted to cases where the SO(3) group acts on the input space, which prevents their use on 2D images. Falorsi et al. (2018) and Park et al. (2022) extract 3D rotational equivariant features from images to model object orientation, but were limited to simplistic datasets with a single object. Similar to our work, Esteves et al. (2019a) learns SO(3) equivariant embeddings from image input for object pose prediction; however, they use a supervised loss to replicate the embeddings of a spherical convolutional network pretrained on 3D images. In contrast, our method incorporates a novel architecture for achieving SO(3) equivariance from image inputs that can be trained end-to-end on the challenging pose prediction tasks.

**Uncertainty over** SO(3)    Due to object symmetry or occlusion, there may be a set of equivalent rotations that result in the same object appearance, which makes pose prediction challenging. Most early works into object pose prediction have avoided this issue by either breaking the symmetry when labelling the data (Xiang et al., 2014) or applying loss functions to handle to known symmetries (Xiang et al., 2018; Wang et al., 2019). However, this approach requires knowing what symmetries are present in the data, and does not work for objects that have ambiguous orientations due to occlusion (e.g. coffee mug when the handle is not visible). Several works have proposed models to reason about orientation uncertainty by predicting the parameters of von Mises (Prokudin et al.,

Figure 1: Illustration of our proposed model, Image2Sphere (I2S). First, output image features of a pre-trained ResNet are orthographically projected to the sphere. We convolve the features $\Psi$ with a learned filter on $S^2$ to generate a signal on $\mathrm{SO}(3)$ (represented by vector field on the sphere). A final $\mathrm{SO}(3)$ group convolution is performed to produce a detailed distribution over 3D rotations, allowing I2S to learn object symmetries and represent uncertainty during pose estimation.

2018), Fisher (Mohlin et al., 2020), and Bingham (Gilitschenski et al., 2019; Deng et al., 2022) distributions. However, multi-modal distributions required for modeling complex object symmetries which can be difficult to train. Recent work by Murphy et al. (2021) leverages an implicit model to produce a non-parametric distribution over $\mathrm{SO}(3)$ that can model objects with large symmetry groups. Our method also generates a distribution over $\mathrm{SO}(3)$ that can model symmetric objects, but uses $\mathrm{SO}(3)$-equivariant layers to achieve higher accuracy and sample efficiency.

## 3 BACKGROUND

### 3.1 EQUIVARIANCE

Equivariance formalizes what it means for a map $f$ to be symmetry preserving. Let $G$ be a symmetry group, that is, a set of transformations that preserves some structure. Assume $G$ acts on spaces $\mathcal{X}$ and $\mathcal{Y}$ via $\mathcal{T}_g$ and $\mathcal{T}'_g$, respectively. For all $g \in G$, a map $f \colon \mathcal{X} \to \mathcal{Y}$ is equivariant to $G$ if

$$f(\mathcal{T}_g x) = \mathcal{T}'_g f(x). \tag{1}$$

That is, if the input of $f$ is transformed by $g$, the output will be transformed in a corresponding way. Invariance is a special case of equivariance when $\mathcal{T}'_g$ is the identity map (i.e. applying group transformations to the input of an invariant mapping does not affect the output).

### 3.2 GROUP CONVOLUTION OVER HOMOGENEOUS SPACES

The group convolution operation is a linear equivariant mapping which can be equipped with trainable parameters and used to build equivariant neural networks (Cohen & Welling, 2016). Convolution is performed by computing the dot product with a filter as it is shifted across the input signal. In standard 2D convolution, the shifting corresponds to a translation in pixel space. Group convolution (Cohen & Welling, 2016) generalizes this idea to arbitrary symmetry groups, with the filter transformed by elements of the group. Let $G$ be a group and $\mathcal{X}$ be a homogeneous space, i.e. a space on which $G$ acts transitively, for example, $\mathcal{X} = G$ or $\mathcal{X} = G/H$ for a subgroup $H$. We compute the group convolution between two functions, $f, \psi \colon \mathcal{X} \to \mathbb{R}^k$, as follows:

$$[f \star \psi](g) = \sum_{x \in \mathcal{X}} f(x) \cdot \psi(\mathcal{T}_g^{-1} x). \tag{2}$$

Note that the output of the convolution operation is defined for each group element $g \in G$, while the inputs, $f$ and $\psi$, are defined over a homogeneous space of $G$. By parameterizing either $f$ or $\psi$ using trainable weights, group convolution may be used as a layer in an equivariant model.

### 3.3 SO(3)–EQUIVARIANCE

For reasoning about physical objects, such as pose detection or part segmentation, it is desirable to learn functions that are equivariant to the group $\mathrm{SO}(3)$ of 3D rotations. In order to build $\mathrm{SO}(3)$-equivariant neural networks using group convolution, it is necessary to efficiently parameterize the space of signals $\{f \colon \mathcal{X} \to \mathbb{R}\}$ over the homogenous spaces $\mathcal{X} = S^2$ or $\mathcal{X} = \mathrm{SO}(3)$. Cohen et al. (2018b) give an effective and compact solution in terms of the truncated Fourier transform. Analogous to the Fourier basis over the circle, it is possible to give a Fourier basis for signals defined over $S^2$

in terms of the spherical harmonics $Y_k^l$ and for signals over $\mathrm{SO}(3)$ in terms of Wigner $D$-matrix coefficients $D_{mn}^l$. Writing $f \colon \mathrm{SO}(3) \to \mathbb{R}$ in terms of the $D_{mn}^l$ and then truncating to a certain frequency $l \leq L$ we obtain an approximate representation $f(g) \approx \sum_{l=0}^{L} \sum_{m=0}^{2l+1} \sum_{n=0}^{2l+1} c_{mn}^l D_{mn}^l(g)$.

$\mathrm{SO}(3)$ group convolution (2) can be efficiently computed in the Fourier domain using the convolution theorem[1]. Namely, the convolution of two functions is calculated as the element-wise product of the functions in the Fourier domain. For functions over $\mathrm{SO}(3)$, the Fourier domain is described by the Wigner $D$-matrix coefficients, corresponding to a block diagonal matrix with a $(2l+1) \times (2l+1)$ block for each $l$ (Knapp, 1996). A functions over $S^2$ in the Fourier domain is described by a vector of coefficients of spherical harmonics and convolution is performed using an outer product (Cohen et al., 2018a). This generates the same block diagonal matrix output as $\mathrm{SO}(3)$ convolution (e.g. both $S^2$ and $\mathrm{SO}(3)$ convolution generate signals that live on $\mathrm{SO}(3)$). See Geiger et al. (2022) for efficient implementation of Fourier and inverse Fourier transforms of $S^2$ and $\mathrm{SO}(3)$ signals.

### 3.4 SO(3) Distributions with the Fourier Basis

The output of the above group convolution is a signal $f \colon \mathrm{SO}(3) \to \mathbb{R}$ defined as a linear combination of Wigner $D$-matrices. While this signal is useful for efficient group convolution, it cannot be directly normalized to produce a probability distribution. Instead, the signal can be queried at discrete points in $\mathrm{SO}(3)$ and then normalized using a softmax. To reduce errors introduced by the discretization, points should be taken from an equivolumetric grid. Equivolumetric grids over $\mathrm{SO}(3)$ can be generated using an extension of the HEALPix method developed by Yershova et al. (2010). The HEALPix method (Gorski et al., 2005) starts with 12 'pixels' that cover the sphere, and recursively divides each pixel into four sections. The latitude and longitude of each pixel describe two Euler angles of a rotation. The extension to $\mathrm{SO}(3)$ specifies the final Euler angle in a way that the grid is equally-spaced in $\mathrm{SO}(3)$. The resolution of the final grid can be controlled by the number of recursions.

## 4 METHOD

Our method, Image2Sphere (I2S), is designed to predict object pose from images under minimal assumptions. Much previous work assumes objects with either no symmetry or known symmetry, and thus it is sufficient to output a single point estimate of the pose. Our method, in contrast, outputs a distribution over poses, which allows us to represent uncertainty due to partial observability and the inherent ambiguity in pose resulting from object symmetries. Recent work using distribution learning has suffered from difficulty in training multi-modal distributions and high data requirements. I2S circumvents these challenges by reasoning about uncertainty in the Fourier basis of $\mathrm{SO}(3)$, which is simple to train over and allows us to leverage equivariant layers for better data efficiency.

The I2S network consists of an encoder, a projection step, and a 2-layer spherical pose predictor (see Figure 1). I2S generates $\mathrm{SO}(3)$ equivariant features from traditional convolutional network encoders. Our method then maps features from image space to the 2-sphere using an orthographic projection operation. Then, we perform two spherical convolution operations. Importantly, the output of the spherical convolution is a signal over the Fourier basis of $\mathrm{SO}(3)$, which can be queried in the spatial domain to create highly expressive distributions over the space of 3D rotations.

### 4.1 Mapping from Image to Sphere

We use orthographic projection to map the output image feature map $f \colon \mathbb{R}^2 \to \mathbb{R}^h$ from the ResNet to a spherical signal $\Psi \colon S^2 \to \mathbb{R}^h$. Orthographic projection $P \colon S^2 \to \mathbb{R}^2$ defined $P(x, y, z) = (x, y)$ maps a hemisphere onto the unit disk. By positioning the feature map $f$ around the unit disk before projecting, we get a localized signal $\Psi(x) = f(P(x))$ supported over one hemisphere. The advantage of this method is that it preserves the spatial information of features in the original image. Practically, to compute $\Psi$, a HEALPix grid is generated over a hemisphere $\{x_i\} \subset S^2$ and mapped to positions $\{P(x_i)\}$ in the image feature map. The value of $\Psi(x_i)$ is interpolated from the value of $f$ at pixels near position $P(x_i)$. This process is illustrated in Figure 2.

---

[1] See Cohen et al. (2018a) for explanation and visuals describing efficient $\mathrm{SO}(3)$ group convolution.

We then move $\Psi$ to the frequency domain using fast Fourier transform to express $\Psi$ as a linear combination of spherical harmonics. Operating in the frequency domain allows us to efficiently convolve continuous signals over the sphere ($S^2$) and the group of 3D rotations (SO(3)). We truncate the Fourier series for $\Psi$ at frequency $L$ as $\Psi(x) \approx \sum_{l=0}^{L} \sum_{k=0}^{2l+1} c_k^l Y_k^l(x)$. This truncation may cause sampling error if the spatial signal has high frequency. We mitigate this error by: (1) tapering the magnitude of the projected features toward the edge of the image to avoid a discontinuity on the 2-sphere, and (2) performing dropout of the HEALPix grid such that a random subset of grid points is used for each projection.

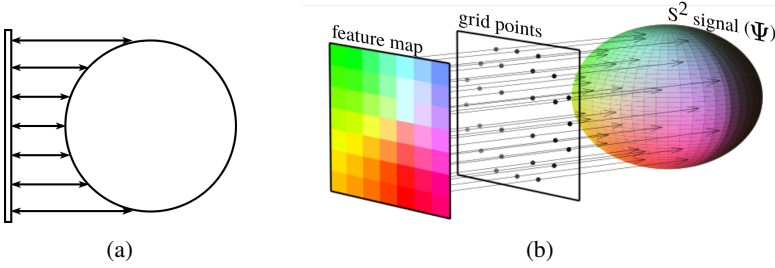

(a)             (b)

Figure 2: Projection of features from image space to the 2-sphere. (a) orthographic projection links pixels in image space to points on the sphere using vectors orthogonal to the image plane; (b) dense feature map, visualized as an RGB image, is mapped to a signal over $S^2$ by orthographically projecting onto all visible grid points, resulting in a signal that is non-zero over half the 2-sphere.

## 4.2 LEARNING SO(3) EQUIVARIANT FEATURES

Once the learned features are projected to the sphere, we use operations that preserve the 3D symmetry of the representation. First, the spherical signal $\Psi$ is processed with an $S^2$-equivariant group convolution operation. Unlike traditional convolutional layers that use locally supported filters, we elect to use a filter with global support. Intuitively, we view this operation similarly to template matching where the projected hemisphere signal $\Psi$ is the template and we are looking for the rotation transformation that results in the best match with the filter. Conveniently, using a globally supported filter gives a global receptive field after one layer, allowing for a shallower network. This is important since each spherical convolution operation is fairly computation and memory intensive due to the high band limit $L$ necessary to model potentially complex distributions on the output.

After the $S^2$-convolution, we perform one SO(3)-equivariant group convolution before outputting the probability density. Using a locally supported filter, this operation serves as a refinement step and results in slightly higher prediction accuracy (see comparison in Appendix C.2). Following Spherical CNN (Cohen et al., 2018a), we apply non-linearities between convolution layers by mapping the signal to the spatial domain, applying a ReLU, and then mapping back to the Fourier domain.

## 4.3 LOSS FUNCTION

The output of the spherical convolution operation is a signal $f \colon \mathrm{SO}(3) \to \mathbb{R}_{\geq 0}$ that indicates the probability that the image corresponds to a particular object orientation. This output is queried using an equivolumetric grid over SO(3) then normalized with a softmax operation to produce a categorical distribution, which can be optimized using a cross-entropy loss. We find that this loss is effective even when training on images with ambiguous orientation or symmetric objects. For these cases, the model learns to predict high probabilities for the set of equivalently likely orientations. Additionally, we find that the model can accurately predict likelihoods for grids of higher resolution than it was trained on (e.g. train with a grid resolution of 5 degrees and test with a resolution of 2.5 degrees).

## 5 EXPERIMENTS

### 5.1 DATASETS

To demonstrate the strengths of our method, we evaluate it on several challenging object orientation estimation datasets. Additional details can be found in Appendix B.2.

The first dataset, **ModelNet10-SO(3)** (Liao et al., 2019), is composed of rendered images of synthetic, untextured objects from ModelNet10 (Wu et al., 2015). The dataset includes 4,899 object instances over 10 categories, with novel camera viewpoints in the test set. Each image is labelled with a single 3D rotation matrix, even though some categories, such as desks and bathtubs, can have an ambiguous pose due to symmetry. For this reason, the dataset presents a challenge to methods that cannot reason about uncertainty over orientation.

Next, **PASCAL3D+**, (Xiang et al., 2014), is a popular benchmark for pose estimation that includes real images of objects from 12 categories. The dataset labels symmetric object categories in a consistent manner (e.g. bottles are symmetric about the z-axis, so label zero rotation about the z-axis), which simplifies the task. To improve accuracy, we follow the common practice of augmenting the training data with synthetic renderings from Su et al. (2015). Nevertheless, PASCAL3D+ still serves as a challenging benchmark due to the high variability of natural textures and presence of novel instances in the test set.

Lastly, the **SYMSOL** dataset was recently introduced by Murphy et al. (2021) to evaluate the expressivity of methods that model distributions over 3D rotations. It includes synthetic renderings of objects split into two groups: geometric objects like the tetrahedron or cylinder with complex symmetries (SYMSOL I), and simple objects with single identifying feature such that the pose is ambiguous when the feature is occluded (SYMSOL II). The dataset provides the full set of equivalent rotation labels for each image so methods can be evaluated on how well they capture the distribution. Note that Murphy et al. (2021) generates results using 100k renderings per shape, which we report as a baseline, but the publicly released dataset that we train on only includes 50k renderings per shape.

### 5.2 EVALUATION METRICS

The goal of pose prediction is to minimize the angular error between the predicted 3D rotation and the ground truth 3D rotation. Two commonly used metrics are the median rotation error (*MedErr*) and the accuracy within a rotation error threshold (e.g. *Acc@15* is the fraction of predictions with 15 degrees or less rotation error). However, these metrics assume that there exists a single, ground truth 3D rotation, which is not valid for symmetric objects or images with pose ambiguity. For ModelNet10-SO(3) and PASCAL3D+, only a single rotation is provided so these metrics must be used. However, when the full set of equivalent rotation labels are provided, like with SYMSOL, a more informative measure is the average log likelihood, computed as the expected log likelihood that the model, $p_\theta$, assigns to rotations sampled from the distribution of equivalent rotations $p_{GT}$: $\mathbb{E}_{R \sim p_{GT}}[\log p_\theta(R|x)]$. Achieving high log likelihood requires modelling all symmetries of an object.

### 5.3 NETWORK AND TRAINING DETAILS

I2S uses a residual network (He et al., 2016) with weights pretrained on ImageNet (Deng et al., 2009) to extract dense feature maps from 2D images. We use a ResNet50 backbone for ModelNet10-SO(3) and SYMSOL, and ResNet101 for PASCAL3D+. The orthographic projection uses a HEALPix grid with recursion level of 2, out of which 20 points are randomly selected during each forward pass. We parameterize the learned $S^2$ filter in the Fourier domain, e.g. learn weights for each spherical harmonic. The filter in the $SO(3)$ convolutional layer is locally supported over rotations up to 22.5 degrees in magnitude. We query the signal in the spatial domain using $SO(3)$ HEALPix grids with 36k points (7.5 degree spacing) during training and 2.4M (1.875 degree spacing) during evaluation. We use the same maximum frequency ($L = 6$) on all datasets. Additional details on the architecture can be found in Appendix B.1.

I2S is instantiated with PyTorch and the `e3nn` library (Geiger et al., 2022). It is trained using SGD with Nesterov momentum of 0.9 for 40 epochs using a batch size of 64. The learning rate starts at 0.001 and decays by factor of 0.1 every 15 epochs.

## 5.4 BASELINES

We compare our method against competitive baselines including regression methods and distribution learning methods. Zhou et al. (2019) and Brégier (2021) predict valid 3D rotation matrices using differentiable orthonormalization processes, Gram-Schmidt and Procrustes, respectively, that preclude discontinuities on the rotation manifold. Liao et al. (2019) and Mahendran et al. (2018) use unique classification-regression losses to predict rotation. Prokudin et al. (2018) represents rotation uncertainty with a mixture of von Mises distributions over each Euler angle, while Mohlin et al. (2020) predicts the parameters for a matrix Fisher distribution. Gilitschenski et al. (2019) and Deng et al. (2022) both predict multi-modal Bingham distributions. Lastly, Murphy et al. (2021) trains an implicit model to generate a non-parametric distribution over 3D rotations. All baselines use the same sized, pretrained ResNet encoders for each experiment. Results are from the original papers when available.

## 5.5 MODELNET10-SO3

We report performance on ModelNet10-SO(3) in Table 1. Our method outperforms all baselines on median error and accuracy at 15 degrees. Moreover, I2S achieves the lowest error on nine out of ten categories, and reports less than 5 degrees of median error on eight out of ten categories (full breakdown in Appendix A.2). Because the dataset includes objects with symmetries, the average median error is heavily influenced by high errors on categories with ambiguous pose. For instance, all methods get at least 90 degree rotation error on bathtubs, since it is hard to tell the front from the back. Thus, regression methods can get stuck between ambiguous object symmetries. In contrast, our method generates a distribution which can capture multiple symmetry modes, even if the correct one cannot be identified. We believe that I2S is more accurate than other distributional methods because its equivariant layers encode the symmetry present in the pose estimation problem.

Table 1: Comparison of pose estimation performance on ModelNet10-SO(3) dataset. Data is averaged over all ten object categories. For I2S, we report mean and standard deviation over five random seeds.

|  | Acc@15↑ | Acc@30↑ | MedErr↓ |
|---|---|---|---|
| Zhou et al. (2019) | 0.251 | 0.504 | 41.1 |
| Brégier (2021) | 0.257 | 0.515 | 39.9 |
| Liao et al. (2019) | 0.357 | 0.583 | 36.5 |
| Deng et al. (2022) | 0.562 | 0.694 | 32.6 |
| Prokudin et al. (2018) | 0.456 | 0.528 | 49.3 |
| Mohlin et al. (2020) | 0.693 | **0.757** | 17.1 |
| Murphy et al. (2021) | 0.719 | 0.735 | 21.5 |
| I2S (ours) | **0.728** ± **0.002** | 0.736±0.002 | **15.7±2.9** |

## 5.6 PASCAL3D+

PASCAL3D+ is a challenging benchmark for pose detection that requires robustness to real textures and generalization to novel object instances. We report the median error of our method and competitive baselines in Table 2. Importantly, our method achieves the lowest median error averaged across all 12 categories. Note that the symmetries are broken consistently in the labels, so outperforming methods that regress to a single pose, e.g. Mahendran et al. (2018), is particularly impressive. We hypothesize that the SO(3) equivariant layers within our model provide stronger generalization capabilities. Example predictions of our method show that it captures reasonable uncertainty about object pose when making predictions (Figure 3).

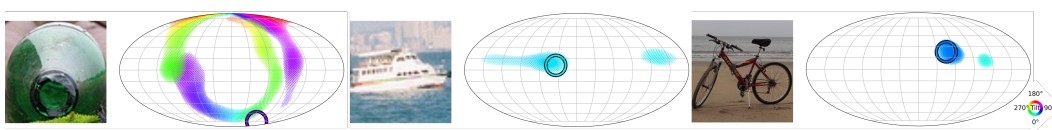

Figure 3: Example predictions of I2S on PASCAL3D+. The distribution visualizations show that our method captures pose uncertainty, e.g. it is unclear if a long boat is approaching or a shorter is headed to the right, creating ambiguity about one rotation axis (middle).

| | avg. | plane | bicycle | boat | bottle | bus | car | chair | table | mbike | sofa | train | tv |
|---|---|---|---|---|---|---|---|---|---|---|---|---|---|
| Zhou et al. (2019) | 19.2 | 24.7 | 18.9 | 54.2 | 11.3 | 8.4 | 9.5 | 19.4 | 14.9 | 22.5 | 17.2 | 11.4 | 17.5 |
| Brégier (2021) | 20.0 | 27.5 | 22.6 | 49.2 | 11.9 | 8.5 | 9.9 | 16.8 | 27.9 | 21.7 | 12.6 | 10.2 | 20.6 |
| Liao et al. (2019) | 13.0 | 13.0 | 16.4 | 29.1 | 10.3 | 4.8 | 6.8 | 11.6 | 12.0 | 17.1 | 12.3 | 8.6 | 14.3 |
| Mohlin et al. (2020) | 11.5 | 10.1 | 15.6 | 24.3 | 7.8 | 3.3 | 5.3 | 13.5 | 12.5 | 12.9 | 13.8 | 7.4 | 11.7 |
| Prokudin et al. (2018) | 12.2 | 9.7 | 15.5 | 45.6 | **5.4** | **2.9** | 4.5 | 13.1 | 12.6 | 11.8 | **9.1** | **4.3** | 12.0 |
| Tulsiani & Malik (2015) | 13.6 | 13.8 | 17.7 | 21.3 | 12.9 | 5.8 | 9.1 | 14.8 | 15.2 | 14.7 | 13.7 | 8.7 | 15.4 |
| Mahendran et al. (2018) | 10.1 | **8.5** | 14.8 | **20.5** | 7.0 | 3.1 | 5.1 | **9.5** | 11.3 | 14.2 | 10.2 | 5.6 | 11.7 |
| Murphy et al. (2021) | 10.3 | 10.8 | 12.9 | 23.4 | 8.8 | 3.4 | 5.3 | 10.0 | **7.3** | 13.6 | 9.5 | 6.4 | 12.3 |
| I2S (ours) | **9.8** | 9.2 | **12.7** | 21.7 | 7.4 | 3.3 | 4.9 | **9.5** | 9.3 | **11.5** | 10.5 | 7.2 | **10.6** |
| | ±0.4 | ±0.4 | ±0.7 | ±1.3 | ±0.7 | ±0.1 | ±0.1 | ±0.8 | ±3.4 | ±0.8 | ±0.8 | ±0.5 | ±0.6 |

Table 2: Median rotation error (°) on PASCAL3D+. First column shows average median error over all twelve classes. For some baselines, we report the corrected results by Murphy et al. (2021) (see Appendix B.2 for details). For I2S, we report mean and standard deviation over six runs.

## 5.7 SYMSOL

One of the strengths of our method is the ability to represent distributions over 3D rotations. As shown in the previous section, this formulation is beneficial when training on symmetric objects. In this section, we quantitatively evaluate the ability to model uncertainty in two settings: SYMSOL I which includes simple geometric objects with complex symmetries and SYMSOL II which has objects marked with a single identifier such that self-occlusion creates pose ambiguity. Because most images correspond to a set of equivalent rotation labels, we measure performance using average log likelihood, reported in Table 3.

The results show that our method effectively represents complex distributions over $SO(3)$. We achieve higher log likelihood on all SYMSOL shapes than methods that map to a specific small family of distributions such as von Mises or Bingham. This highlights the advantage of parametrizing uncertainty in the Fourier basis of $SO(3)$, which is a simpler approach that avoids training multi-modal distributions. We note that Murphy et al. (2021) outperforms our method when trained on 100k images per object; however, our method is better when trained on only 10k images per object. This demonstrates an important distinction between the two approaches: our method explicitly encodes the 3D rotation symmetry of the problem in the spherical convolutions, whereas Murphy et al. (2021) must learn the symmetry from data. We argue that sample efficiency is important for real world applications since it is not practical to collect 100k images of an single object to model its symmetry. We want to highlight that we use the same maximum frequency $L$ for all experiments in this work which shows that I2S can be deployed without knowing if object symmetries are present in the task.

Table 3: Average log likelihood on SYMSOL datasets. SYMSOL I includes objects with complex symmetries, while SYMSOL II includes objects whose poses can be ambiguous under self-occlusion. The highest likelihood in each column is in bold, second-best is underlined.

| num. training images | | | | SYMSOL I | | | | | SYMSOL II | | |
|---|---|---|---|---|---|---|---|---|---|---|---|
| | | avg. | cone | cyl. | tet. | cube | ico. | avg. | sphX | cylO | tetX |
| 100k | Deng et al. (2022) | -1.48 | 0.16 | -0.95 | 0.27 | -4.44 | -2.45 | 2.57 | 1.12 | 2.99 | 3.61 |
| | Gilitschenski et al. (2019) | -0.43 | 3.84 | 0.88 | -2.29 | -2.29 | -2.29 | 3.70 | 3.32 | 4.88 | 2.90 |
| | Prokudin et al. (2018) | -1.87 | -3.34 | -1.28 | -1.86 | -0.50 | -2.39 | 0.48 | -4.19 | 4.16 | 1.48 |
| | Murphy et al. (2021) | **4.10** | **4.45** | **4.26** | **5.70** | **4.81** | 1.28 | **7.57** | **7.30** | **6.91** | **8.49** |
| | I2S (ours) | 3.41 | 3.75 | 3.10 | 4.78 | 3.27 | **2.15** | 4.84 | 3.74 | 5.18 | 5.61 |
| 10k | Murphy et al. (2021) | -7.94 | -1.51 | -2.92 | -6.90 | -10.04 | -18.34 | -0.73 | -2.51 | 2.02 | -1.70 |
| | I2S (ours) | **2.98** | **3.51** | **2.88** | **3.62** | **2.94** | **1.94** | **3.61** | **3.12** | **3.87** | **3.84** |

## 5.8 COMPARISON OF ALTERNATIVE IMAGE TO SO(3) MAPPINGS

We argue that a main driver of our method's pose accuracy is $SO(3)$-equivariant processing. While many existing methods for end-to-end $SO(3)$-equivariance have been explored in the literature (Fuchs et al., 2020; Deng et al., 2021; Cohen et al., 2018a), it is not well-understood how to combine non-equivariant and equivariant layers in a network. In this section, we consider other sensible approaches to map from features in the image plane to features that live on $SO(3)$ or a discrete subgroup of it. The approach taken by I2S is to perform orthographic projection to link features on the sphere to the image plane (spatial projection), then convolve it with a filter that is parametrized

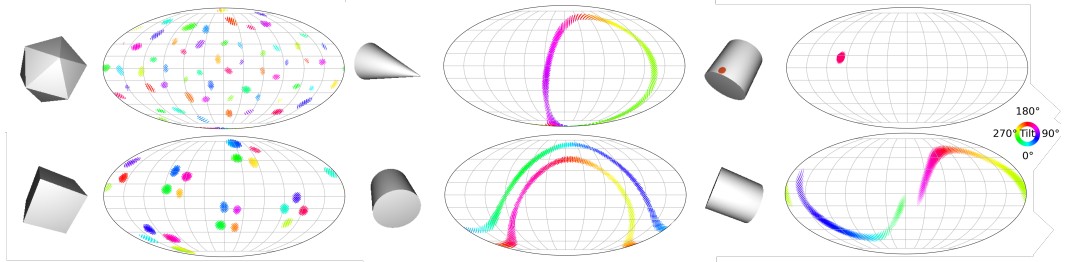

Figure 4: Distributions generated by I2S for SYMSOL objects. To visualize the probabilities over $SO(3)$, rotations with non-negligible probability are plotted as dots using a Mollweide projection, encoding rotation orthogonal to the sphere with color (Murphy et al., 2021). I2S is able to capture large discrete symmetry groups of the cube and icosahedron (left column), as well as continuous symmetries of the cone and cylinder (middle column). I2S predicts single pose when the dot is visible on cylinder (*cylO*), but is uncertain when the dot is occluded (right column).

by the Fourier basis of $S^2$ to generate a signal over $SO(3)$ (Fourier filter). Sensible alternatives can be proposed by instead linking features in the image to coefficients of the Fourier basis (Fourier projection) or parametrizing the filter with features over the sphere (spatial filter). Alternatively, one may consider the approach of directly computing the coefficients of the Fourier basis using an MLP. Finally, we include a modification of the I2S framework using the icosahedral group convolution (a discrete group of $SO(3)$) instead of spherical convolution. This model uses an anchor-regression framework to produce continuous rotation predictions, as outlined in Chen et al. (2021).

Table 4: Comparing different approaches to learn an $SO(3)$ signal with image inputs. Results show orientation prediction performance on ModelNet10-SO(3) with limited training views.

|  | *Acc@15* | *Acc@30* | *MedErr* |
|---|---|---|---|
| I2S* (spatial projection, Fourier filter) | **0.623** | 0.640 | 46.3 |
| I2S (spatial projection, spatial filter) | 0.610 | **0.633** | **44.6** |
| I2S (Fourier projection, spatial filter) | 0.523 | 0.533 | 56.8 |
| I2S (Fourier projection, Fourier filter) | 0.511 | 0.631 | 57.0 |
| Fourier basis w/ MLP | 0.360 | 0.390 | 69.2 |
| Icosahedral group version | 0.455 | 0.582 | 57.6 |

We evaluate the pose prediction performance of these alternative approaches in Table 4 on the ModelNet10-SO(3) dataset with limited training views. I2S in all its variations, outperforms using an MLP to parametrize the Fourier basis and the discrete icosahedral group convolution version. This is likely because pose estimation benefits from equivariance to the continuous rotation group. We find that using a spatial projection is important for accuracy, which we hypothesize is better able to preserve spatial information encoded in the dense feature map. How the filter is parametrized is less influential. We use the Fourier basis to parametrize the filter, while the original spherical CNN work parametrizes the filter using a spatial grid over the sphere (Cohen et al., 2018a).

## 6 CONCLUSION

In this work, we present the first method to leverage $SO(3)$-equivariance for predicting distributions over 3D rotations from single images. Our method is better suited than regression methods at handling unknown object symmetries, generates more expressive distributions than methods using parametric families of multi-modal distributions while requiring fewer samples than an implicit modeling approach. We demonstrate state-of-the-art performance on the challenging PASCAL3D+ dataset composed of real images. One limitation of our work is that we use a high maximum frequency, $L$, in the spherical convolution operations to have higher resolution predictions. Because the number of operations in a spherical convolution is quadratic in $L$, it may be impractical for applications where more spherical convolutions are required.

ACKNOWLEDGMENTS

This work is supported in part by NSF 1724257, NSF 1724191, NSF 1763878, NSF 1750649, and NASA 80NSSC19K1474. R. Walters is supported by the Roux Institute and the Harold Alfond Foundation and NSF grants 2107256 and 2134178.

REPRODUCIBILITY STATEMENT

The code to replicate the results of our method is available at `https://github.com/dmklee/image2sphere`. All datasets used are publicly available and a thorough description of preprocessing is provided in Appendix B.2. The model architecture and training protocol are discussed in Section 5.3 and Appendix B.1. Information on the baselines and their reported results is provided in Appendix B.3.

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

# A    DETAILED RESULTS

## A.1    MODELNET10-SO3 LIMITED TRAINING SET

We compare the sample efficiency of pose prediction methods by training on ModelNet10-SO(3) with limited training views in Table 5. Specifically, we reduce the number of training views from 100 per instance to 20 per instance. Our method outperforms the baselines by a larger margin in the low-data setting. This results highlights an important distinction between our method and the baselines: our method explicitly encodes the 3D rotation symmetry present in the pose prediction problem, whereas other methods must learn the symmetry from data.

Table 5: Comparison on ModelNet10-SO(3) with limited training views. Methods are trained on five times fewer images than the experiment in Table 1.

|  | Limited Training Set | | |
|---|---|---|---|
|  | *Acc@15↑* | *Acc@30↑* | *MedErr↓* |
| Zhou et al. (2019) | 0.064 | 0.239 | 62.7 |
| Brégier (2021) | 0.129 | 0.359 | 51.5 |
| Murphy et al. (2021) | 0.515 | 0.533 | 59.5 |
| I2S (ours) | **0.623** | **0.640** | **46.3** |

## A.2    MODELNET10-SO3 PER-CLASS BREAKDOWN

Table 6 reports the median rotation error for each object in ModelNet10-SO(3). These results highlight the challenge of pose prediction with unknown symmetries. Note that the categories that are hard to predict, e.g. bathtub, night stand and table, can have multiple correct reference frames due to symmetry.

Table 6: Per-class median rotation error ($°$) on ModelNet10-SO(3). Results for I2S are reported as mean and standard deviation over six random seeds.

|  | avg. | bathtub | bed | chair | desk | dresser | monitor | n. stand | sofa | table | toilet |
|---|---|---|---|---|---|---|---|---|---|---|---|
| Zhou et al. (2019) | 41.1 | 103.3 | 18.1 | 18.3 | 51.5 | 32.2 | 19.7 | 48.4 | 17.0 | 88.2 | 13.8 |
| Brégier (2021) | 39.9 | 98.9 | 17.4 | 18.0 | 50.0 | 31.5 | 18.7 | 46.5 | 17.4 | 86.7 | 14.2 |
| Liao et al. (2019) | 36.5 | 113.3 | 13.3 | 13.7 | 39.2 | 26.9 | 16.4 | 44.2 | 12.0 | 74.8 | 10.9 |
| Deng et al. (2022) | 32.6 | 147.8 | 9.2 | 8.3 | 25.0 | 11.9 | 9.8 | 36.9 | 10.0 | 58.6 | 8.5 |
| Mohlin et al. (2020) | 17.1 | **89.1** | 4.4 | 5.2 | 13.0 | 6.3 | 5.8 | 13.5 | 4.0 | 25.8 | 4.0 |
| Prokudin et al. (2018) | 49.3 | 122.8 | 3.6 | 9.6 | 117.2 | 29.9 | 6.7 | 73.0 | 10.4 | 115.5 | 4.1 |
| Murphy et al. (2021) | 21.5 | 161.0 | 4.4 | 5.5 | 7.1 | 5.5 | 5.7 | 7.5 | 4.1 | 9.0 | 4.8 |
| I2S (ours) | **16.3** | 124.7 | **3.1** | **4.4** | **4.7** | **3.4** | **4.4** | **4.1** | **3.0** | **7.7** | **3.6** |
|  | ±2.9 | ±28.1 | ±0.0 | ±0.0 | ±0.1 | ±0.1 | ±0.1 | ±0.1 | ±0.0 | ±1.0 | ±0.1 |

## A.3 VISUALIZING PREDICTIONS

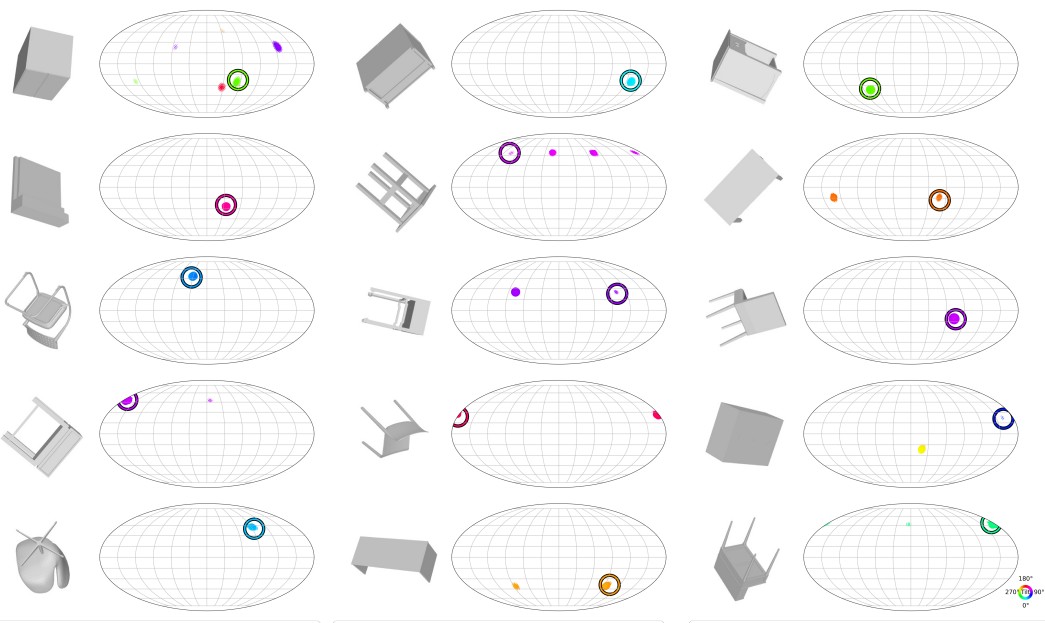

Figure 5: Example predictions of our method on ModelNet10-SO(3) with 100 training views. The dataset includes objects with two-fold (i.e bathtub) or four-fold symmetry (i.e. side table), which our method can model as multi modal distributions in SO(3).

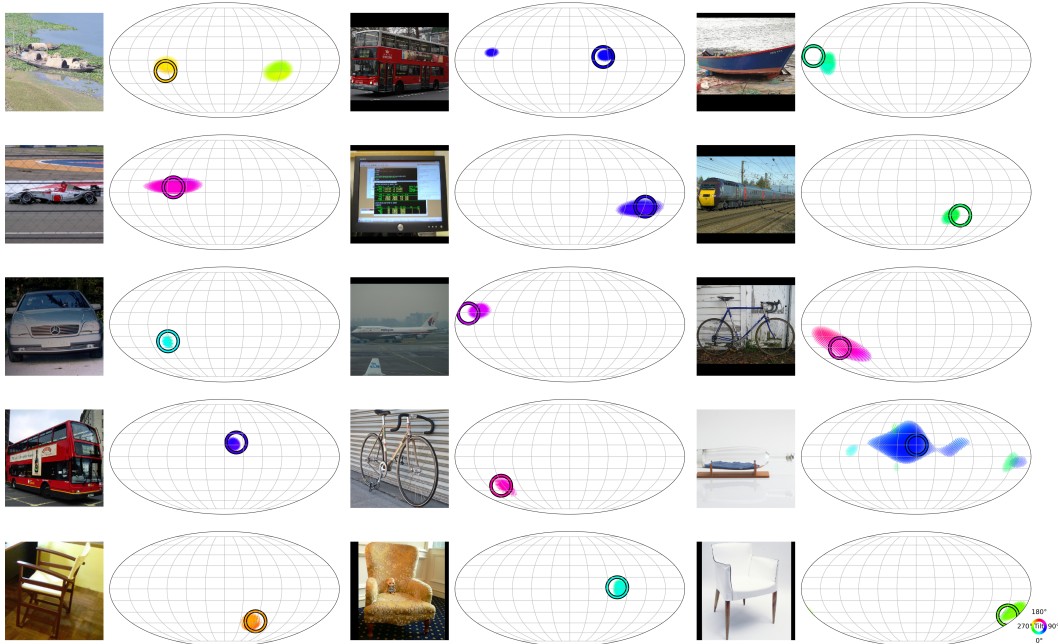

Figure 6: Additional predictions of our method on PASCAL3D+. Ground truth label is shown as circular ring. The distribution over SO(3) is visualized by converting to Euler angles: the first two are represented spatially via Mollweide projection and the third is encoded as color. Our model captures uncertainty in the pose that result from object symmetries or insufficient visual cues.

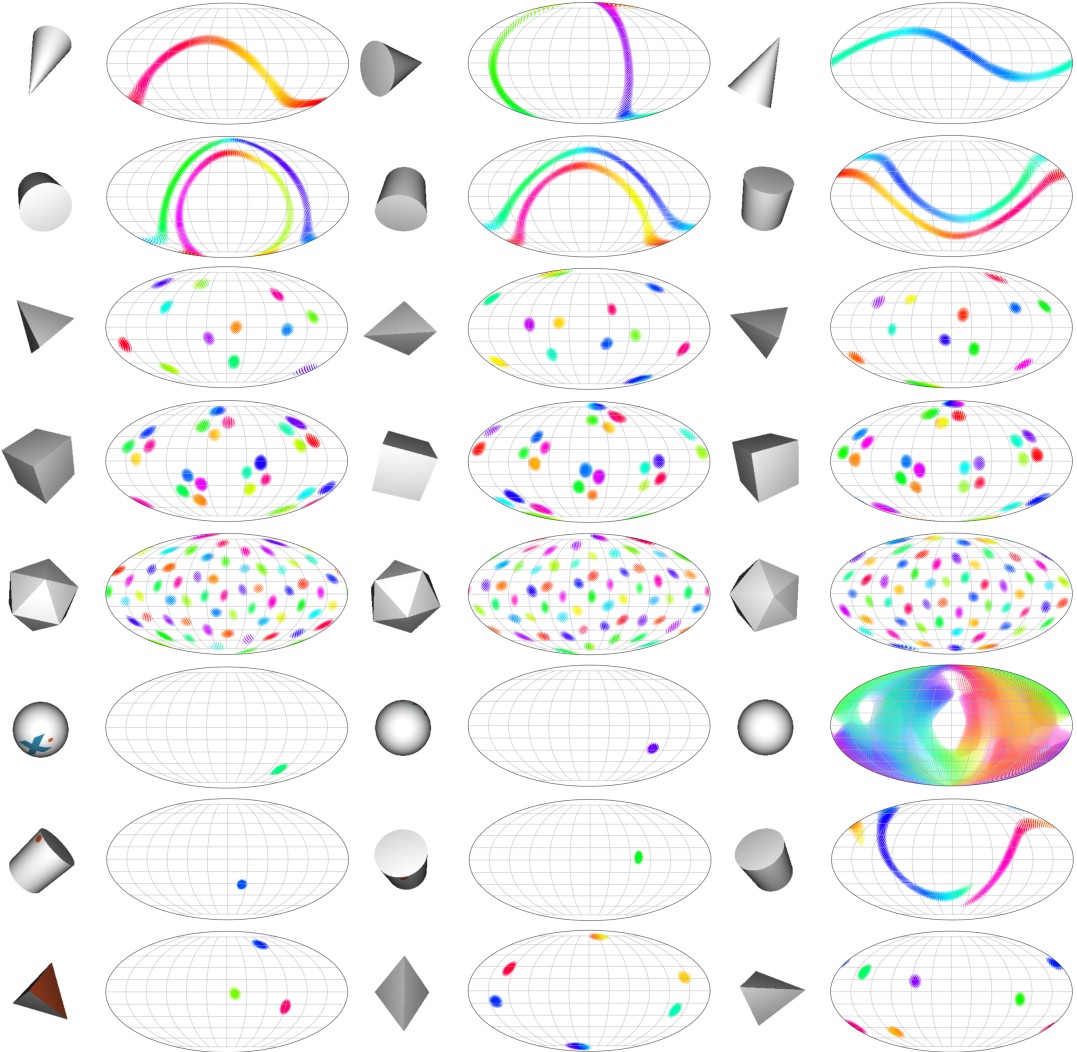

Figure 7: I2S capturing object symmetries and pose ambiguity on SYMSOL Dataset (50k views). Each row corresponds to a different shape from SYMSOL: cone, cylinder, tetrahedron, cube, icosahedron, circleX, cylinderO, and tetrahedronX. I2S is able to represent complex object symmetries, both discrete and continuous.

# B IMPLEMENTATION DETAILS

## B.1 ARCHITECTURE

We use a ResNet encoder with weights pretrained on ImageNet. With 224x224 images as input, this generates a 7x7 featuremap with 2048 channels. The orthographic projection onto the sphere is performed using a HEALPix grid of recursion level 2 restricted to half the sphere. With each forward pass, 20 of these grid points are randomly sampled and used to generate the $S^2$ signal. The $S^2$ signal is converted to the Fourier domain with a maximum frequency of 6. A spherical convolution operation is performed using a filter that is parametrized in the Fourier domain, which generates an 8-channel signal over SO(3). A non-linearity is applied by mapping the signal to the spatial domain, applying a ReLU, then mapping back to Fourier domain. One final spherical convolution with a locally supported filter is performed to generate a one-dimensional signal on SO(3). The signal is queried using an SO(3) HEALPix grid (recursion level 3 during training, 5 during evaluation) and then normalized using a softmax. The network is instantiated in PyTorch, and we use the `e3nn`[2] library for the group convolution operations.

## B.2 DATASET PREPARATION

**ModelNet10-SO(3)** is available for download at the Github[3] associated with Liao et al. (2019). The dataset has a standardized train and test split. It provides two training sets: one with 100 views per object instance (we call this the Full Training Set), and one with 20 views per object instance (we call this the Limited Training Set). The test set has 4 views per instance. Each image is labeled with a single rotation label, and object symmetries were not broken during labeling. Based on the original work, we do not perform any data augmentation with this dataset.

**SYMSOL** can be downloaded from the Github[4] linked by Murphy et al. (2021). The dataset includes renderings of 8 synthetic objects split into two categories: SYMSOL I includes tetrahedron, cube, icosahedron, cone and cylinder, while SYMSOL II includes marked tetrahedron, marked cylinder, and marked sphere. For each shape, there are 50k renderings in the training set and 5K renderings in the test set. Each image is labeled with the set of valid rotations (for continuous symmetries like the cylinder, it is provided at 1 degree increments). During training, the set of valid rotations is randomly sampled to generate a single rotation label to compute the loss. In Table 3, we use results originally reported by Murphy et al. (2021), and follow the approach of training a single model on all objects from SYMSOL I but different models for each object in SYMSOL II. The results reported by Murphy et al. (2021) were generated using 100k training renderings per shape, but the publicly released dataset only has 50k. Thus, this strongly favors the baselines.

**Pascal3D+** is available for download at the link[5] provided in Xiang et al. (2014). The training data is found in the ImageNet_train, ImageNet_val, and PASCALVOC_train folders, and the test data is in PASCALVOC_val. Following Murphy et al. (2021), we discard any data that is labeled *occluded*, *difficult* or *truncated*. We follow the data augmentation procedure from Mohlin et al. (2020) that randomly performs horizontal flip and slight perspective transformation during training. Additionally, we supplement the training data with synthetic images from RenderForCNN (Su et al., 2015) (this requires free ImageNet account to download), such that three quarters of data is synthetic during training. Including synthetic data is important to achieve high accuracy, and is also used by the baselines. Where possible, we use the reported results of the baselines on PASCAL3D+; however, as pointed out by Murphy et al. (2021), some of the baselines used incorrect evaluation functions or a different evaluation set. In these cases, we report the performance from Murphy et al. (2021), since they re-ran these baselines after correcting the issues.

---

[2] `https://e3nn.org`
[3] `https://github.com/leoshine/Spherical_Regression`
[4] `https://github.com/google-research/google-research/tree/master/implicit_pdf`
[5] `https://cvgl.stanford.edu/projects/pascal3d.html`

### B.3 BASELINES

Here, we include details for baselines that we implemented ourselves. For information on other baselines, refer directly to their work.

**Zhou et al. (2019)** proposes using the Gram-Schmidt process to convert a 6D vector into a valid, 3x3 rotation matrix. To implement this network, we used a ResNet architecture with spatial pooling on the final feature map. The resulting vector was processed with two linear layers to generate the 6D vector. The model is trained with an L2 loss function using a ground truth rotation matrix. The method is trained on all classes at once, but uses a separate linear layer to predict the rotation of each class.

**Brégier (2021)** proposes using the Procrustes method to convert a 9D vector into a valid, 3x3 rotation matrix. They provide an efficient implementation of the Procrustes method. The architecture and loss function is the same as for Zhou et al. (2019), except the linear layer produces a 9D vector.

The original work of **Liao et al. (2019)** only showed results for PASCAL3D+, which used an incorrect evaluation function as noted by Murphy et al. (2021). Thus, for PASCAL3D+, we used the results reported by Murphy et al. (2021) with the corrected evaluation function. For ModelNet10-SO(3), we ran their code using a pretrained ResNet50 as an encoder.

### B.4 CREATING VISUALS

We follow the visualization code that was publicly released by Murphy et al. (2021) to represent distributions over 3D rotations as a 2D plot. The probability associated with each rotation is represented as a dot, with the size proportional the magnitude. The rotation is encoded by converting to XYX euler angles, the first two angles correspond to latitude and longitude in a Mollweide projection and the final angle is encoded as color using an HSV colormap. To make the visualizations more interpretable, we do not plot any probabilities less than a given threshold. In our visualizations, we generate probabilities associated with the $SO(3)$ HEALPix grid used during model evaluation (recursion level of 5; 2.4M points).

## C ADDITIONAL EXPERIMENTS

### C.1 EFFECT OF MAXIMUM FREQUENCY $L$

To efficiently perform $SO(3)$ group convolutions, we must restrict our representations to in the frequency domain. Learning with lower frequency signals can be more efficient and may generalize better in some cases, while higher frequency signals may be better for encoding complex distributions like those shown in Figure 4. In Table 7, we perform a small experiment showing the effects of different maximum frequencies, $L$, on pose prediction accuracy. Note that we use $L = 6$ for all other experiments in this work. Interestingly, we find that including higher frequencies in the representation, e.g. $L > 6$ can actually reduce accuracy, despite the additional learnable parameters.

Table 7: Varying maximum frequency $L$ in Fourier basis. Results are generated on ModelNet10-SO(3) with limited training views. The same $L$ is maintained through both spherical convolutions of our method.

|  | *Acc@15*↑ | *Acc@30*↑ | *MedErr*↓ |
|---|---|---|---|
| $L = 2$ | 0.560 | 0.656 | 33.6 |
| $L = 4$ | 0.626 | 0.646 | 46.6 |
| $L = 6$ | 0.623 | 0.640 | 46.3 |
| $L = 8$ | 0.605 | 0.621 | 43.7 |
| $L = 10$ | 0.599 | 0.618 | 46.8 |

## C.2 Effect of Additional SO(3) Convolutions

Table 8: Effect of $SO(3)$ convolutional layers on ModelNet10-SO(3) pose prediction. I2S, as proposed, includes one $SO(3)$ convolution following the $S^2$ convolution. We show performance without the $SO(3)$ convolution, and with an additional $SO(3)$ convolution. Additional $SO(3)$ convolutions provide minimal benefit at the expense of compute.

| | Full Training Set | | | Limited Training Set | | |
| --- | --- | --- | --- | --- | --- | --- |
| | *Acc@15↑* | *Acc@30↑* | *MedErr↓* | *Acc@15↑* | *Acc@30↑* | *MedErr↓* |
| I2S, no $SO(3)$ conv | 0.729 | 0.737 | 15.6 | 0.616 | 0.634 | 46.4 |
| I2S, one $SO(3)$ conv | 0.729 | 0.737 | 14.4 | 0.623 | 0.640 | 46.3 |
| I2S, two $SO(3)$ conv | 0.729 | 0.737 | 19.0 | 0.625 | 0.643 | 45.5 |

Our method performs one $S^2$ convolution followed by one $SO(3)$ convolution. In this section, we look at the performance benefits of this final refinement layer and potential benefits of an additional convolution. We find that the performing $SO(3)$ convolution does lead to a marginal improvement, with little to be gained by performing more. This suggests that most of the $SO(3)$ reasoning occurs within the first spherical convolution that uses a learned filter with global support.

