# OpenReview forum: "Image to Sphere: Learning Equivariant Features for Efficient Pose Prediction"
_ICLR.cc/2023/Conference — ICLR 2023 notable top 5%_

### Official Review · Reviewer_PnmT · 2022-10-18

**Confidence:** 4
**Correctness:** 3
**Technical Novelty And Significance:** 3
**Empirical Novelty And Significance:** 3
**Recommendation:** 8

**Clarity, Quality, Novelty And Reproducibility:**

The paper is generally clear except what I wrote above and I believe that the experiments are reproducible in principle. I do think that the paper may not be novel enough for acceptance, but I hope the authors can convince me otherwise.

**Details Of Ethics Concerns:**

No ethics concerns.

**Strength And Weaknesses:**

Strengths:

1. The proposed architecture and loss function -- I2S -- are simple and intuitive.

2. The paper is well-written with good illustrations and visualizations of results for important cases that show the usefulness of the I2S and that I2S works predictably for some known objects, which is important.

3. Experiments on multiple datasets show that I2S indeed outperforms all the baselines considered almost always.

Weaknesses:

1. Novelty of the method: Given that both $SO(3)$-equivariant convolutions as well as pose prediction by predicting distributions over $SO(3)$ are existing ideas, I am not sure that the architecture in this paper is novel enough.

2. Comparison with Murphy (2021): Experiments on SYMSOL indicate that Murphy (2021) outperforms I2S when the entire dataset is available. However, I2S performs better in a few-shot setting. I think the authors should try to clarify why exactly mapping to the sphere and then the SO(3) is important, this seems to be the main difference from Murphy (2021), which is not clear in the current version of the paper.

3. Experimental results should have error bars and statistical significance values.



**Summary Of The Paper:**

The paper describes a novel SO(3) equivariant architecture for pose prediction of objects from 2D images. The neural network takes image features from an image encoder and maps it to a sphere, which is followed by an $S^2$ convolution and one or more $SO(3)$ convolutions. Both these layers are equivariant to $SO(3)$ transformations. An important feature is that the output of the neural network is a distribution on the $SO(3)$ pose manifold. This feature is important in the case where the objects have symmetries and the pose can be known only up to that symmetry. Experiments on multiple pose prediction datasets show improved performance over baselines as well as the ability to produce distributions that show understanding of object symmetries.

**Summary Of The Review:**

Overall, I think the paper studies an important problem in computer vision, and proposes a simple and effective architecture, which takes into account symmetries in the objects for pose prediction. I do think some of the experimental results and novelty of the paper have to be explained and done better. I am inclining towards acceptance.

UPDATE AFTER AUTHOR RESPONSE:

I appreciate that the authors have responded too all my comments. I think the paper is in good shape to be accepted although architectural novelty is limited. It does show how to use existing spherical and SO(3) convolutional layers for an important problem and shows some improvements in performance. I am increasing my rating based on the authors' response to my comments and those of the other reviewers.

---

> ### Author Response · Authors · 2022-11-18
> **Response to Reviewer PnmT**
>
> > Given that both SO(3)-equivariant convolutions as well as pose prediction by predicting distributions over SO(3) are existing ideas, I am not sure that the architecture in this paper is novel enough
>
> Our method is novel relative to existing work in two ways.  First, while SO(3)-equivariant networks are well-studied, our method processes 2D images, a space in which the group action is not well-defined.  Our work is one of only a few papers to investigate neural networks with non-end-to-end equivariance properties, and the only one to benchmark on challenging datasets with real-world images.  In this way, our method extends the benefits of SO(3)-equivariant processing to applications that use 2D input modalities.  Second, ours is the first deep learning method, to our knowledge, to use the Fourier basis of SO(3) to parametrize distributions over SO(3).  Our approach is simpler to optimize and more expressive than methods that use multi-modal distributions, and empirically outperforms an implicit method at modeling object symmetries in the low-data regime (see Table 3).
>
> > Comparison with Murphy (2021): Experiments on SYMSOL indicate that Murphy (2021) outperforms I2S when the entire dataset is available. However, I2S performs better in a few-shot setting. I think the authors should try to clarify why exactly mapping to the sphere and then the SO(3) is important, this seems to be the main difference from Murphy (2021), which is not clear in the current version of the paper.
>
> By mapping to the sphere and performing SO(3) group convolutions, our method encodes symmetry to 3D rotations of its representation.  The explicit symmetry encoded in our network improves generalization, which explains why our method outperforms Murphy (2021) in the few-shot setting.  In Table 4, we show that mapping to the sphere is more effective than mapping directly to the spherical harmonics; intuitively, this is because mapping to the sphere preserves spatial information in the image.  We also ablate our method by substituting linear layers for the group convolution layers which reduces Acc@15 from 0.623 to 0.360.  This demonstrates that the SO(3) group convolutions are important to the success of our method.  In the updated version, we include more experiments (see Table 3 and 5) that show our method outperforms Murphy et al. (2021) in the few-shot setting.
>
> > Experimental results should have error bars and statistical significance values
>
> We thank the reviewer for this suggestion.   We added standard deviation values for our method in the updated version of the paper (we evaluate ModelNet10-SO(3) with five seeds and PASCAL3D+ with six seeds).  We do not include this information for baselines since they did not report results for multiple seeds.

---

### Official Review · Reviewer_CAH5 · 2022-10-23

**Confidence:** 2
**Correctness:** 4
**Technical Novelty And Significance:** 3
**Empirical Novelty And Significance:** 4
**Recommendation:** 8

**Clarity, Quality, Novelty And Reproducibility:**

Clarity: The paper is well-written and easy to read.

Quality: The equality is high, with a novel design and solid evaluation.

Novelty: The proposed method is novel.

Reproducibility: The source code is provided and there should no issues in reproducing.


**Strength And Weaknesses:**

Strength:
1. The paper proposed a novel orientation distribution estimation method from simple image features. The idea of mapping image features to a sphere and further a 3D manifold is very interesting.
2. SO(3)-equivariant Group convolution is devised for rotation estimation on the manifold. This brings a new idea to the field and I think it is inspiring and could be further extended to other applications.
3. The paper is well-written and easy to follow.

Weakness:
1. The evaluation is only conducted on images with a single object, well cropped and centered in the images.  I am wondering how the method performs in other more cluttered and occluded datasets (e.g. YCB-V and LM-O). Learned or ground truth detection/segmentation may be used for this purpose.
2. I would like to hear the authors’ comments on how well the proposed method to unseen object instances/categories.


**Summary Of The Paper:**

This paper proposes Image2Sphere a rotation distribution estimation method by mapping 2D image features to the 3D rotation manifold and applying SO(3) equivariant convolution on the 3D rotation manifold. This is done by first mapping the image features onto a sphere using orthographic projection, and then a learned filter is applied to transform the signal onto SO(3). Finally, SO(3) group convolution is performed to produce the distribution over the 3D rotation space. Thanks to the SO(3) equivariant group convolution operation, the pose estimation distribution can be modeled well in the results. The method is evaluated on several image-based orientation estimation results and the proposed method is reported to outperform prior methods and other baselines, especially in a low data regime.

**Summary Of The Review:**

The paper proposes a novel method for the distributional estimation of SO(3) rotation. The evaluation is solid and shows the efficacy of the method. I tend to accept this paper, yet the paper does not fully align with my background and I may not have a full understanding of the field.

---

> ### Author Response · Authors · 2022-11-18
> **Response to Reviewer CAH5**
>
> > I am wondering how the method performs in other more cluttered and occluded datasets
>
> We agree this is an interesting experiment to perform, but do not have time to finish it in the rebuttal period.  In future work, we plan to integrate our method into a 6D pose estimation framework so it can be applied to datasets like YCB-V and LM-O.
>
> > I would like to hear the authors’ comments on how well the proposed method to unseen object instances/categories
>
> Our method generalizes to novel instances.  We achieve state-of-the-art performance on the PASCAL3D+ dataset, which uses unseen instances in the test set.  We would argue that orientation prediction on novel object categories is an ill-posed problem. The model outputs the orientation of the object as a 3D rotation relative to some unrotated version of the object.  If the model has been trained on labeled objects in the same category, it is reasonable for it to infer the unrotated orientation of a novel object and thus get the correct answer.  However, for an object in a novel category, the unrotated reference state is essentially undefined.

---

### Official Review · Reviewer_9pzm · 2022-10-25

**Confidence:** 3
**Correctness:** 3
**Technical Novelty And Significance:** 1
**Empirical Novelty And Significance:** 2
**Recommendation:** 6

**Clarity, Quality, Novelty And Reproducibility:**

Novelty: As I commented, the idea seems not novel to me.

Clarity: More clarifications are needed to describe the proposed neural network.

Quality: The overall quality of this work is ok.

**Strength And Weaknesses:**

Strength:

1. The paper aims to solve an interesting and challenging problem: estimating 3-D pose from 2-D images, which has broad applications in 3-D computer vision and 3-D imaging.

Weaknesses and questions:

1. Although the writing is in general ok, there are some missing details in the implementation. I encourage authors to provide a more detailed description of each layer and operations in the proposed neural network. For example, the authors did not clarify how exactly equation (2) is implemented in the Fourier space of SO(3).

2. The idea of this work is not sufficiently motivated. I can see the advantages of learning the distribution of SO(3) can mitigate the issue of symmetry. However, it remains unclear how the S2 convolution and the operations shown in Figure 2 are essential to the success of the proposed method. More explainations and experiments are expected to demonstrate the effectiveness of each module of the proposed network.

3. The idea of this work is not novel. It seems to me that proposed method is a combination of ideas from previous works such as group convolution,  Fourier expansion on SO(3) and etc. Learning distribution on SO(3) to break symmetry is also not a new idea, see [1].

4. The proposed method is not the only work that predicts rotations from 2-D images, so there are some missing references. See [2] that uses each 2-D image to predict the pose of protein molecule for 3-D reconstruction. I also encourage the authors to have some discussions that compare this work with your method, since they are able to obtain reliable pose estimates from 2-D images using a simple neural network.

[1] Synchronizing Probability Measures on Rotations via Optimal Transport.

[2] CryoAI: Amortized Inference of Poses for Ab Initio Reconstruction of 3D Molecular Volumes from Real Cryo-EM Images

**Summary Of The Paper:**

This paper proposes a deep neural network to predict poses in SO(3) from 2-D images. The proposed method first maps 2-D images onto a hemi-sphere by projection, and then convolve it with a filter on $S^2$ in the basis of spherical harmonics, and then apply rotation-equivariant layers that convolves the signals in the basis of Wigner D-matrices. The proposed method appears to be more effective than other networks on a variety of datasets in handling ambiguity that is caused by object symmetry.

**Summary Of The Review:**

In summary, this is an ok submission. However, there is a large room for improvement in terms of clarity and experiments. The idea of this work is neither novel nor sufficiently motivated.

---

> ### Author Response · Authors · 2022-11-18
> **Response to Reviewer 9pzm**
>
> > There are some missing details in the implementation
>
> We updated the writing to better explain how Equation 2 is implemented. We include more details on individual network layers in Appendix B.1.
>
> > The idea of this work is not sufficiently motivated. I can see the advantages of learning the distribution of SO(3) can mitigate the issue of symmetry. However, it remains unclear how the S2 convolution and the operations shown in Figure 2 are essential to the success of the proposed method. More explanations and experiments are expected to demonstrate the effectiveness of each module of the proposed network.
>
> We believe our design is well motivated, but we agree this could be more clearly explained in the paper.   The S$^2$ and SO(3) convolutions encode the inherent 3D rotation symmetry of pose prediction.  In Table 4, we justify the proposed architecture with an ablation study.  Substituting group-equivariant convolution layers with linear layers reduces performance significantly, from 0.623 Acc@15 to 0.360 on ModelNet10-SO(3) Limited.  We also find that the spatial projection to S$^2$ is more effective than mapping directly to the spherical harmonics; our intuition is that the spatial projection preserves the positional information present in the image.   We will improve the writing in the methods section to make it more clear why we made certain design decisions.
>
> > It seems to me that the proposed method is a combination of ideas from previous works such as group convolution, Fourier expansion on SO(3), etc.  Learning distribution on SO(3) to break symmetry is also not a new idea, see [1].
>
> Our method incorporates SO(3) group convolution with non-equivariant layers in a novel way that distinguishes it from existing work.  Existing work uses SO(3) group convolution in an end-to-end manner, meaning the SO(3) group action is present in the input (i.e. point cloud data or spherical images).  In contrast, I2S performs SO(3) equivariant processing after extracting features from 2D images, meaning the symmetry is partially learned during training.  From a distributional learning perspective, unlike other work, I2S parametrizes the distribution over SO(3) using the Fourier basis.  Our parametrization is simpler and more expressive than existing multi-modal approaches, and can learn with fewer data than an implicit model (see Table 3).
>
> > I also encourage the authors to have some discussions that compare this work with your method, since they are able to obtain reliable pose estimates from 2-D images using a simple neural network.
>
> Thank you for pointing out this work.  The cryoAI work uses a similar method to one of our baselines which we show our method I2S outperforms. Namely, it uses a convolutional encoder to predict the rotation parameters of an image using the Gram-Schmidt process from Zhou et al. [1].  It could be interesting to see if cryoAI could be improved using our method of rotation prediction.  Although the differences in domain would require significant modifications to our method, this would be an interesting direction for future work.  We include a mention of cryoAI in the related works section in the updated version.
>
> [1] Zhou, Yi, et al. "On the continuity of rotation representations in neural networks." Proceedings of the IEEE/CVF Conference on Computer Vision and Pattern Recognition. 2019.

---

> > ### Comment · Reviewer_9pzm · 2022-11-25
> > **Response to the Rebuttal**
> >
> > Thanks for the detailed explanation, and it addressed all of my concerns. Therefore, I increased my score to 6.

---

### Official Review · Reviewer_n1bx · 2022-10-30

**Confidence:** 4
**Correctness:** 4
**Technical Novelty And Significance:** 3
**Empirical Novelty And Significance:** Not applicable
**Recommendation:** 8

**Clarity, Quality, Novelty And Reproducibility:**

The paper has good quality since it addresses the problem of object symmetries using equivariant layers and allows the representation of uncertainty. Some details are scarce, namely those concerning the computation of group convolution in Fourier space. The main contribution of the paper is the ability to provide uncertainty estimates for the poses of symmetric objects.

**Details Of Ethics Concerns:**

There are no ethics concerns.

**Strength And Weaknesses:**

The strengths include the ability to handle unknown object symmetries and the generation of distributions over poses, which in addition of allowing to take into account the pose ambiguity from object symmetries, it permits the representation of uncertainties due to partial observability. However with some datasets (SYMSOL) previous methods (Murphy et al.) have better performance. Computational cost not evaluated.

**Summary Of The Paper:**

This paper describes an approach for pose estimation for objects with symmetries. The approach leverages SO(3)-equivariance to predict distributions over 3D rotations from single images. Image features are obtained from a pre-trained ResNet. These features are orthographically projected to the sphere where the features are convolved with a learned filter on S^2. As a result a signal is generated on SO(3). A group convolution on SO(3) is performed generating a detailed distribution over SO(3). This distribution can model symmetric objects. As a result object symmetries are learned and uncertainty can be represented.

**Summary Of The Review:**

This is a paper describing an approach for pose estimation of symmetric objects with prediction of uncertainty. The contribution is clear but incremental.

---

> ### Author Response · Authors · 2022-11-18
> **Response to Reviewer n1bx**
>
> > with some datasets (SYMSOL) previous methods (Murphy et al.) have better performance.
>
> In the SYMSOL experiment, we report results in a small and big data regime.  In the big-data regime, the previous method does better.  However, we would like to highlight the results in the small-data regime, where our method significantly outperforms Murphy et al.  We argue that the performance in the small-data regime is more practical, since viewing a single object from 100K viewpoints is not usually practical.  We performed more experiments in the small data regime for SYMSOL II and ModelNet10-SO(3), which confirm that our method is more effective than Murphy et al. in the small data regime.
>
>
> Average log likelihood (higher is better) on SYMSOL II dataset with 10k training views per object (Table 3):
> .| Average | sphX | cylO | tetX
> :---------|-------------|--------|--------|-------
> Murphy et al. (2021) | -0.73 | -2.51 | 2.02 | -1.70
> I2S (ours) | 3.61 | 3.12 | 3.87 | 3.84
>
> Comparison on ModelNet10-SO(3) Limited (five times fewer training samples than original) (Table 5):
> . | Acc@15$\uparrow$ | Acc@30$\uparrow$ | MedErr $\downarrow$
> :---|---|---|---
> Murphy et al. (2021) | 0.515 | 0.533 | 59.5
> I2S (ours) | 0.623 | 0.640 | 46.3
>
> $\newline$
> > Computational cost not evaluated
>
> The computational cost of training our model is comparable to regression methods like Liao et al. [1].  It takes 47 minutes per epoch to train our model on ModelNet10-SO(3) using a GeForce GTX 1080 Ti (compare this to 49 min/epoch to train Liao et al.).
>
> > Some details are scarce, namely those concerning the computation of group convolution in Fourier space.
>
> We appreciate the feedback on this.  In the updated version, we added more details to our explanation of the group convolution in Fourier space and include a footnote directing interested readers to the original Spherical CNN work.
>
> [1] Liao, Shuai, Efstratios Gavves, and Cees GM Snoek. "Spherical regression: Learning viewpoints, surface normals and 3d rotations on n-spheres." Proceedings of the IEEE/CVF Conference on Computer Vision and Pattern Recognition. 2019.

---

### Author Response · Authors · 2022-11-18
**Main Response**

We thank the reviewers for their insightful comments and helpful suggestions.  We are happy to see that most reviewers agreed that our method effectively handles pose uncertainty due to object symmetries, and outperforms other methods on several pose prediction benchmarks.  In the revised version, we improve the explanation for how group convolution is performed in the Fourier domain, and further justify our design decision to include S$^2$ and SO(3) convolutions.  We also add two new experiments to highlight our method’s sample efficiency: SYMSOL II at 10k views (Table 3) and ModelNet10-SO(3) with Limited training views (Table 5).

---

### Decision · Program_Chairs · 2023-01-20

**Decision:**

Accept: notable-top-5%

**Justification For Why Not Higher Score:**

N/A

**Justification For Why Not Lower Score:**

The problem is important and the method is well acknowledged by all the reviewers.

**Metareview: Summary, Strengths And Weaknesses:**

The paper studies SO(3) rotation estimation from a single image and addresses symmetry and uncertainty by predicting a distribution in SO(3) with equivariant representation. State-of-the-art performance is presented on PASCAL 3D+. Although there are some clarity issues for the exposition and some complaint on novelty of architecture, all reviewers are positive about the work.

**Note From Pc:**

if the above contains the word "oral" or "spotlight" please see: "oral" presentation means -> notable-top-5% and "spotlight" means -> notable-top-25%. As stated in our emails, we are disassociating presentation type from AC recommendations